# Setting Goals to Reduce Cardiovascular Risk: A Retrospective Chart Review of a Pharmacist-Led Initiative in the Workplace

**DOI:** 10.3390/ijerph20010846

**Published:** 2023-01-02

**Authors:** Alicia E. Klaassen, Anita I. Kapanen, Peter J. Zed, Annalijn I. Conklin

**Affiliations:** 1Faculty of Pharmaceutical Sciences, The University of British Columbia, Vancouver, BC V6T 1Z3, Canada; 2Department of Emergency Medicine, Faculty of Medicine, The University of British Columbia, Vancouver, BC V5Z 1M9, Canada; 3Centre for Health Evaluation and Outcome Sciences (CHÉOS), St. Paul’s Hospital, Vancouver, BC V6Z IY6, Canada

**Keywords:** primary prevention, health promotion, heart disease risk factors, hypertension, risk reduction behavior, professional-patient relations, patient care planning

## Abstract

**Background:** Cardiovascular diseases (CVD) are the second leading cause of death in Canada with many modifiable risk factors. Pharmacists at a Canadian university delivered a novel CVD risk management program, which included goal-setting and medication management. **Aim:** This study aimed to describe what CVD prevention goals are composed of in a workplace CVD risk reduction program, and how might these goals change over time. **Methods:** A longitudinal, descriptive qualitative study using a retrospective chart review of clinical care plans for 15 patients enrolled in a CVD prevention program. Data across 6 visits were extracted from charts (n = 5413 words) recorded from May 2019–November 2020 and analyzed using quantitative content analysis and descriptive statistics. **Results:** Behavioural goals were most popular among patients and were more likely to change over the 12-month follow-up period, compared to health measure goals. Behavioural goals included goals around diet, physical activity (PA), smoking, medication, sleep and alcohol; health measure goals centered on weight measures, blood pressure (BP) and blood lipid levels. The most common behavioural goals set by patients were for diet (n = 11) and PA (n = 9). Over time, goals around PA, medication, alcohol and weight were adapted while others were added (e.g. diet) and some only continued. Patients experienced a number of barriers to their goal(s) which informed how they adapted their goal(s). These included environmental limitations (including COVID-19) and work-related time constraints. **Conclusions:** This study found CVD goal-setting in the pharmacist-led workplace wellness program was complex and evolved over time, with goals added and/or adapted. More detailed qualitative research could provide further insights into the patient-provider goal-setting experience in workplace CVD prevention.

## 1. Introduction

Cardiovascular diseases (CVD) remain the second major cause of death in Canada, and the leading cause of death in women; these affected 53,704 Canadians in 2020 [1]. Moreover, CVD carries a heavy economic burden, with costs associated with CVD in Canada previously being estimated at more than $22.2 billion per year [2]. Canadian employers feel this burden, with employee CVD leading to a reduction in productivity, and an increase in absenteeism and short-term disability [3]. Therefore, some employers have started to offer workplace CVD screening and risk reduction programs. These CVD programs have been shown to improve employee risk scores, quality of life, reduce medical spending and reduce absenteeism [4,5,6].

A novel pharmacist-led workplace wellness program in Western Canada was recently evaluated and showed positive results. The program was run by a university-affiliated, licensed, pharmacist-led health site with a focus on patient care, learner education, and development of pharmacist practice models in the Canadian context [7], in partnership with the university’s Health, Wellbeing and Benefits team. The pharmacist-led and employer-supported health and wellness program included 178 patients who completed a 12-month intervention, which resulted in significant improvements in clinical markers, risk scores, quality of life and medication adherence [8,9].

The key interventions of the program focused on patient action and behavioural change, with the pharmacist providing structured support and motivational reinforcement. CVD risk management requires addressing many lifestyle factors, including smoking cessation, diet, physical activity, alcohol consumption and medication use. Previous studies have shown that motivational interviewing and goal-setting can improve a patient’s success in making these lifestyle changes [5,8,9,10,11,12,13,14,15,16,17]. However, patient or provider perspectives on the goal-setting process have not been well outlined in cardiovascular disease prevention. Only one paper was identified in CVD self-management and the focus in this case was on management of current disease and not the prevention of future disease; specifically, Brown et al. focused on collaborative goal-setting in a sample of 30 older men managing chronic hypertension [18]. Using a semi-structured interview approach, that study identified that patients lacked specific goals and plans for self-management and highlighted the need for future studies to improve supports for effective goal-setting. Other research on perspectives around goal-setting have focused primarily on rehabilitation and self-management in other disease conditions such as diabetes and asthma [19,20,21,22,23,24,25,26,27,28,29]. These studies concluded that patients preferred a personalized and collaborative approach but also that patients often were unclear about their role in this process.

In collaborative goal-setting, some patients and providers may differ in their prioritization of CVD prevention outcomes and their preferred approach to accomplishing specific CVD risk goals. This means that providers of workplace wellness programs need to adapt the goal-setting process to address a patient’s needs and self-efficacy. Moreover, patient goals may also change over time and in response to various facilitators and barriers [30]. A better understanding of the goal-setting process will allow providers to better adapt their discussions and maintain a patient-centered approach. Leveraging a second implementation of the pharmacist-led program, this descriptive qualitative study aimed to describe which CVD prevention goals are prioritized by patients and whether and how these goals change over time. Our primary research questions were: (1) what do CVD prevention goals and motivation comprise from a patient perspective? and (2) how might goals and motivation change (or not) over time?

## 2. Methods

### 2.1. Study Design and Patient Population

A descriptive qualitative study using retrospective chart review was performed on patient care plans from 15 patients who were enrolled between May–September 2019 for the second phase of the pharmacist-led CVD prevention program which was conducted over 12 months. Patients from the original program were invited (n = 178) to receive a CVD check-in assessment to determine eligibility (i.e., still at-risk of CVD) and willingness to participate in our second evaluation (Appendix A). Patient inclusion criteria were: (i) must be university staff or faculty; (ii) have participated in the pharmacist-led program in 2015–2017; (iii) be enrolled as a primary member in the university extended health benefit plan; and (iv) have a Framingham Risk Score (FRS) above 10% and modifiable risk factors for CVD identified at their most recent check-in assessment [31,32]. Exclusion criteria included: patients under 18 years old and any eligible patients who could not complete the follow-up visits such as those entering retirement or going abroad for work assignment during the study period. Each eligible patient who enrolled would engage in a one-on-one discussion of their CVD health with a pharmacist, using the SMART technique to set out their goal/s [33]. They would then receive up to 5 additional program visits over a 12-month period. Goal-setting questions were developed by a staff pharmacist and goal-related data were recorded by pharmacists during each patient visit. A 10-level ruler was included to measure the level of confidence of a patient in achieving their set goal and was based on the previously validated Readiness Ruler for behavior change [34]. A template outlining these goal-setting questions is given in Appendix A. 

### 2.2. Data Collection

Clinical and demographic data were collected by research staff (AEK, AIK) from patient charts recorded in the electronic medical record system called OSCAR by the pharmacist at each patient visit. Pharmacists’ information on patient characteristics included: age, sex, family history of cardiovascular disease, occupation, measured anthropometry (weight, height, waist circumference, body mass index (BMI)), blood pressure (BP), blood lipids (LDL, HDL, triglycerides) and a calculated Framingham Risk Score (FRS) [31,32]. BMI was used to classify patients as underweight (<18.5 kg/m^2^), normal weight (18.5–24.9 kg/m^2^), overweight (25–29.9 kg/m^2^), and obese (≥30 kg/m^2^). Patients’ BP measurements were described as hypertensive or normotensive based on Hypertension Canada’s 2020 Guideline (using Automatic Office BP Measurement targets, though multiple tests would have been needed to confirm diagnosis of hypertension) [35]. Baseline clinical information was not available in some patients for height and weight (n = 2), BMI (n = 1), BP (n = 1), and triglycerides (n = 8). The lack of triglycerides could be due to the fact that participants were not asked to fast prior to the baseline appointment, and their triglyceride values may have been outside the range for the clinic’s instrument (CholesTect) to detect. Height and weight could be missing as this information was being collected by another provider elsewhere.

Descriptive qualitative data (n = 5413 words) summarizing each clinical encounter collected information on patient goals for CVD prevention, how they planned to achieve these goals, their confidence in their ability to achieve these goals, barriers and challenges to achieving these goals, and their timeline for achieving their goals.

### 2.3. Analytic Approach

Descriptive qualitative data on patients’ goals for CVD prevention (clinical and/or behavioural) were extracted using an inductive coding approach and analysed using quantitative content analysis [36]. This approach was chosen as it allowed for a systematic description of the content of the patient-provider dyad discussions in the clinical encounters, and the quantification of these qualitative data for better tracking of goal changes over time. Information was read and re-read multiple times by the first author for data familiarisation, who also developed initial codes and themes. Data were read and coded line-by-line to develop a codebook containing the list of data-driven categories and their definitions (examples from the data); the codebook was iteratively re-organised until consensus was reached among the research team and categories reflected the research objective of describing what types of goals patients prioritise for CVD prevention and which goals patients maintain or change over time. Quantitative content analysis consisted of multiple phases of data immersion/familiarisation, coding across the dataset, developing and re-organising categories to produce provisional (sub)themes, counting occurrences of each theme, and analysing the qualitative content using descriptive statistics and graphical display. Data extraction and analysis were performed on complete cases using in Microsoft Excel (Microsoft, 2021). Results of the content analysis of qualitative data were reported as means ± standard deviations (SD), median ± interquartile range (IQR), or frequencies (percentages) in tables and figures. We used a bubble plot to display the frequency of change and the type of change in patient behavioural goals throughout the CVD prevention program for every goal category.

## 3. Results

The mean age of patients was 57 years old (SD 5.7) with a range of 45 years to 66 years old. The majority were male (87%, n = 13/15) and 50% of patients had a BMI in the obese category (≥30 kg/m^2^); however, none had a BMI over 40 kg/m^2^ (range: 22.9 to 37.8 kg/m^2^). Forty percent (n = 6/15) of patients had a family history of CVD. On average, patients had a slightly elevated BP reading (median 129/84.5 mmHg ± [17/6]) and a median high-risk waist circumference of 99 cm (IQR = 18) in both males (98.9 cm [14.5]) and females (98.6 cm [NA]). Patients had a range of FRS from 11.7% to 30% with a mean of 18.3 (SD 5.8) which indicated a moderate level of cardiac risk. Over a quarter (27%, n = 4/15) of patients were in the high level of cardiac risk (FRS ≥ 20%). Sample characteristics are given in Table 1 below.

### 3.1. Patient CVD Prevention Goals: Initial Priorities

At baseline, patient goals covered a range of both health behaviours and clinical health measures, with health behaviours being the most expressed goal type for CVD risk reduction (Table 2). Among the 15 patients, common behavioural goals included improving diet (73%) or PA (60%), reducing alcohol intake (27%), taking medication (13%), stopping smoking (13%), and getting better sleep (7%). Many patients focused their health measure goals on losing weight (40%), or reducing blood pressure (33%); some aimed to improve lipid levels (27%). Whereas most patients set only one clinical health measure goal, all patients set multiple goals for health behaviours (Figure 1). Overall, patients expressed a high degree of confidence in achieving their goals, with two-thirds (n = 10/15) reporting their confidence as 7.5 out of 10 (10 = complete confidence, 0 = not at all).

Dietary goals were the most complex as these involved changing quantity and improving quality and/or altering the timing of consumption in a variety of ways (Table 2). Changes to PA typically involved increasing the quantity of exercise but could also include amending the timing of exercise from leisure-based to transport-based. As shown in Table 2, most health behaviour goals were subdivided into several different specific subgoals that patients and providers identified as priorities for their CVD risk reduction. For example, subgoals for better sleep quality included going to bed earlier and avoiding liquids later at night. The diversity of these subgoals was most evident for dietary goals related to changing food quantity. Notably, two-thirds of patients indicated they aimed to increase their vegetable intake and half indicated they aimed to increase their diet’s overall quality.

### 3.2. Patient CVD Prevention Goals over Time

Throughout the 12-month follow-up period, many changes (continued, added and/or adapted) to patients’ goals and subgoals were noted (Figure 2). Most patients (n = 12/15) experienced some change in their goals over 12 months. Goals were never outright halted, though in one instance their discussion was deprioritized due to personal factors (e.g., death of a loved one). All patients who set health measure goals for blood lipids and BP continued these goals throughout follow-up, whereas more patients added or adapted their weight measure goals than patients who continued initial weight goals. Only 4 patients set health measure goals and these were not consistently tracked during follow-up. All the behavioural goals were continued throughout the program and their progress was clearly monitored by the pharmacist with each visit.

Notably, several behavioural goals were added and/or adapted with the exception of smoking cessation (Figure 3). Some patients adapted their goal for reducing alcohol intake, and others added this goal during the program. Goals were also adapted for medication use, but medication use also required adding new goals over time as medication needs changed. Continuation of dietary goals was highly prevalent in the patients’ charts over time, particularly goals focused on changing the quantity of food items (more of some, less of others). In addition to continuation of initial dietary goals, patients often gradually added to existing dietary goals as they became habituated to each dietary (sub)goal. For example, as one patient continued to eat more orange vegetables and less white bread, the pharmacist suggested adding a goal of more purple vegetables and incorporating more of the principles of the Mediterranean diet into their meals which involved both quantitative and qualitative dietary subgoals. The most variation over time was observed for PA-related goals. Although PA goals for exercise duration typically remained the same over time for most patients, about half of these patients also had to adapt their PA goals; several patients also added new PA subgoals throughout the CVD prevention program.

### 3.3. Barriers to Patient Goals for CVD Prevention

Patients also identified barriers they anticipated and/or experienced during the second phase of the CVD prevention program. Several barriers limited patients’ abilities to achieve their behavioural goals and typically required them to make changes; however, barriers were not reported to affect health measure goals. General categories of barriers reported are listed in Table 3 below which includes examples extracted from the summary notes of clinic visit. Major barriers to behavioural goals included limitations related to the physical body, the external environment, work-related time constraints, and psychological factors.

The onset of the COVID-19 pandemic had a significant effect on changes made to CVD prevention goals during follow-up. The goal type most likely to be affected by this external factor related to PA, however goals around alcohol and diet were also affected. In some cases, the pandemic appeared beneficial as patients described being able to increase the duration or frequency of exercise and having fewer opportunities to drink alcohol. Patients who reported the pandemic having a benefit to their PA, had also reported time as a barrier to their CVD prevention goals. Although the pandemic helped some patients find more capacity to exercise while working from home, the pandemic was also seen as a detriment to PA goals for patients who reported they could no longer access the gym. Overall, the COVID-19 pandemic acted as a physical external barrier to patients’ behavioural goals for CVD prevention.

## 4. Discussion

This descriptive, longitudinal qualitative study using retrospective chart review assessed the goal-setting and shared decision-making of patients with pharmacists in a second dose of a 12-month pharmacist-led CVD risk management (prevention) program in the workplace. Overall, we found that health measure goals were the least common and were generally kept unchanged throughout follow-up. By contrast, health behaviour goals were the most popular CVD prevention goals, particularly those related to diet and PA. Diet and PA were also the most complex goals in terms of the large variation in subgoals and the number of changes that were made over time (continuation, adaptation and addition). We also identified important barriers to CVD prevention goals relating to physical limitations, psychological, time and external factors.

### 4.1. Relationship to Previous Work

A recent statement from the American Heart Association outlined the benefits of long-term health behaviour change programs in primary care and community settings for the prevention of CVD [37]. This statement also outlined how progress to implementation of these programs in traditional primary care settings has been slow and varied. This is partially due to a lack of willingness and time, as well as limited opportunity for primary care providers to have these discussions with other healthcare providers [38,39] The workplace offers a new setting where CVD primary prevention can be undertaken with other healthcare providers such as pharmacists using a holistic approach which can best support the efficacy and maintenance of health promotion and CVD risk reduction [5]. While many studies support the role of goal-setting discussions and motivational interviewing in CVD prevention, there is scant literature on the goal-setting process in shared decision-making between patients and providers [5,8,12,17,37,40].

This study of a pharmacist-led workplace wellness initiative at a university found that each behavioural goal, except smoking cessation, required that additions and/or modifications be made over time; this was most notable for diet and PA goals. Most patients held multiple health goals addressing multiple lifestyle factors in the same visit. Other studies support the setting of multiple goals at once, and this can even be beneficial as success with one goal will increase the likelihood of success with another [41,42]. The most common dietary goal that remained unchanged concerned food quantity, typically increasing vegetables in the diet, while dietary goals involving timing or quality of food consumption were frequently added or modified. The continuation of the unchanged dietary goals indicates that they were straightforward, and patients found them both achievable and fitting with their needs. Diet goals were evidence-based and their frequency was similar to that identified in previous cross-sectional and longitudinal studies [40,43,44] however previous data on how these subgoals may change is limited. Physical activity was the most adapted goal type, with increased frequency and/or duration as the most adapted subgoals. This follows from previous research where patients have had difficulty maintaining and achieving PA change, particularly when beginning from more sedentary lifestyles (like our cohort) [45]. As all participants’ job titles represented sedentary occupations, PA goals had more emphasis on getting moving on work days. Other goals did not show this level of complexity, however this could in part be due to a lack of patient interest in taking on these health behaviour changes. Changes in health measure goals were not as consistently recorded although this may be due to deprioritization by patients or by clinicians, or lack of health measure data to discuss. The fact that the discussion was focused on patients’ prioritized health goals does indicate that patient-centered care in this context involves flexibility on the part of the provider, redirecting set goals as needed to address barriers and changing patient priorities.

Barriers expressed by patients fell into external (physical, environmental, time) and internal (psychological) as has been seen in previous studies [46,47,48,49,50]. Lack of social support has been previously reported as a barrier but this was not seen in this study [50]. There was a high incidence of external environmental barriers (particularly for physical activity goals) and this can be attributed to the COVID-19 pandemic, which limited the resources and spaces available to many people in the last few years. Most adaptations were prompted by the worsening of barriers or the appearance of a new barrier. This highlights the need to track patient barriers while they are being followed for health behaviour change.

### 4.2. Methodological Considerations

Patient-pharmacist discussions were tracked using the visit summaries recorded in the clinic EMR (OSCAR). This means that while this study is useful for better understanding clinician-recorded priorities in goal-setting and goal-setting discussions between clinicians and patients, more in-depth details on patients’ perspectives are still needed. In other words, the words in OSCAR do not reflect the direct understanding of the patient and may differ from the patient’s own description and prioritization of their goals. This can be seen in how providers would not always report whether a goal was completed. Goals instead were either presumed continued or deprioritized for that visit. Behavioural change is a continuous and personal process, and so future studies would benefit from patient interview data as well to provide a more complete picture of the process of goal-setting in CVD prevention. Another limitation of this study was the timing during the pandemic. Visits were not held in person at the onset of the COVID-19 pandemic and so health measurements were not able to be collected at the final visit. Thus, only baseline values have been included where available. For the reasons above, goal outcomes and program efficacy were not the focus of this descriptive qualitative study. Multiple different providers took part in the study which provided some heterogeneity in how data were recorded. This second pharmacist-led CVD prevention program from a single Canadian university site is not designed for estimating prevalence in the population and does not include many females.

Nevertheless, this study has some unique strengths. Patients were followed in the “real-life” setting and their regimens were guided and built around their own abilities, preferences and priorities. This meant the study had a very patient-centered approach which could be more easily transitioned to other primary care sites or workplace wellness programs in the future. Retention of patients was also high, with only 1 patient (6%) lost to follow-up and 1 (6%) having withdrawn during the study period. The duration of follow-up for the study was also beneficial as studies in the setting of workplace wellness or goal-setting do not often follow patients for longer than 6 months [5]. This longer duration helps to bolster the “real-life” setting of this study and allowed us to follow patients’ natural variation in motivation, barriers, adherence and goal priorities over the course of a year.

## 5. Conclusions

Pharmacists have a role as healthcare providers in workplace wellness programs targeted at cardiovascular risk reduction for employees. The results of our study have implications for pharmacists continuing to explore these opportunities for clinical service implementation in the workplace. Goal-setting with healthcare providers in this workplace-based pharmacy clinic was a fluid and complex process, with discussion focused on patients’ preferred goals. This could indicate that patient-centered care in this context involves flexibility on the part of the provider, redirecting as needed to address barriers and changing patient priorities. Findings suggest there is scope for additional insights from the patient perspective on CVD risk reduction through further qualitative research such as patient interviews. Additional research is also needed from the provider perspective to further explore this understudied area of goal-setting so as to optimize CVD risk prevention in the working adult population.

## Figures and Tables

**Figure 1 ijerph-20-00846-f001:**
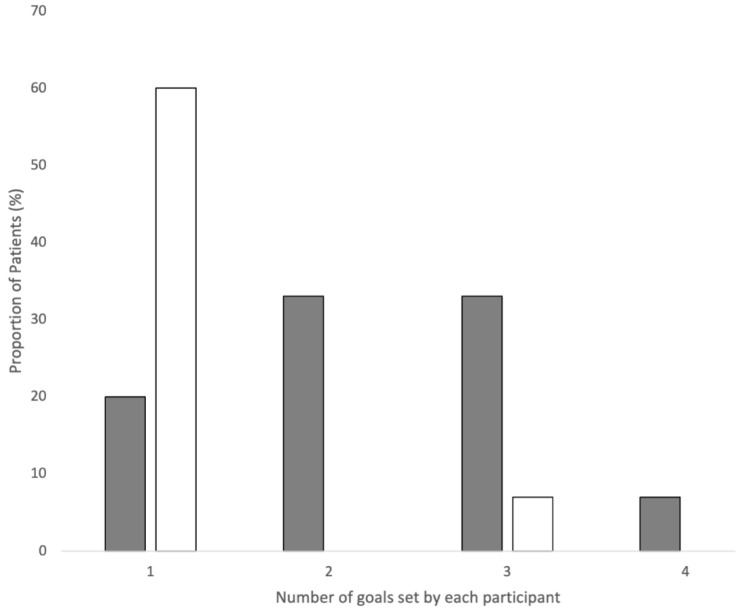
Frequency of patients setting one to four initial CVD prevention goals. Grey, behavioural goals; white, health measure goals.

**Figure 2 ijerph-20-00846-f002:**
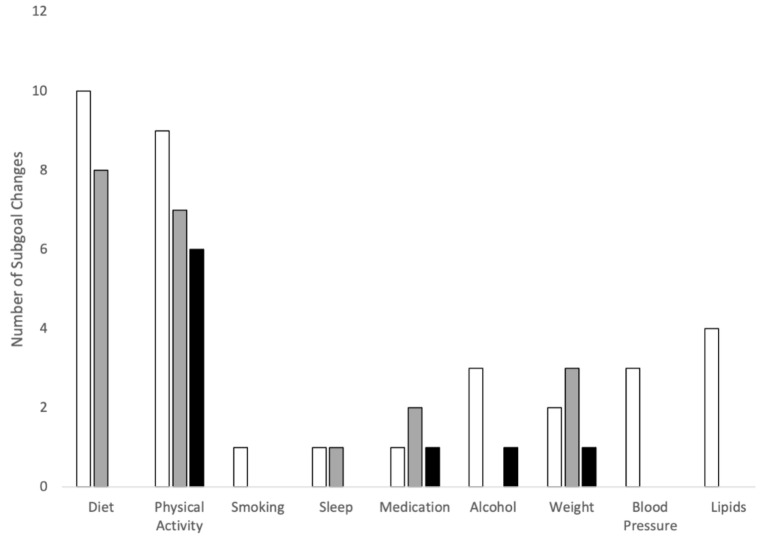
Frequency of subgoal changes in each initial goal for CVD prevention over 12-month follow-up. White, subgoals were continued; grey, subgoals were added; black, subgoals were adapted.

**Figure 3 ijerph-20-00846-f003:**
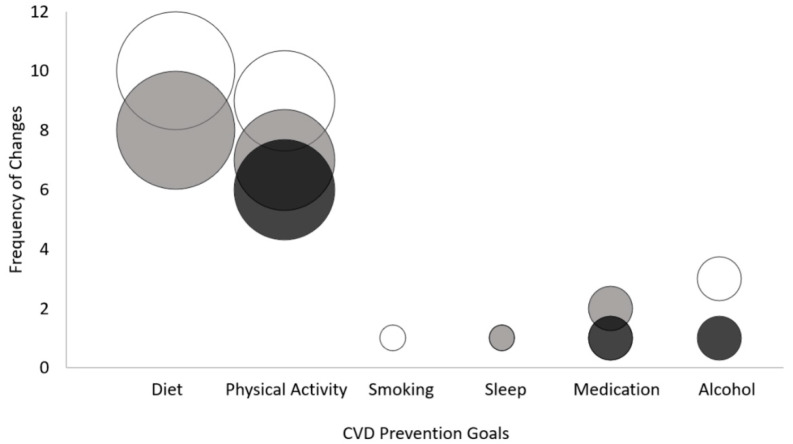
Bubble plot of the frequency of change and the type of change in patient behavioural goals throughout the CVD prevention program for every goal category. PA: physical activity. The *x*-axis lists the broader goal categories. The *y*-axis indicates the frequency of the subgoals within each category which experienced each change. The size of the bubbles indicates the popularity of the goal among patients. White, subgoals were continued; grey, subgoals were added; black, subgoals were adapted.

**Table 1 ijerph-20-00846-t001:** Characteristics of patients in the second pharmacist-led CVD prevention program.

Patient Characteristic	Frequency (%)/Mean (SD)/Median (IQR)
Age	57 years (5.7)
Male	13 (87%)
Family History of CVD	6 (40%)
Job title	
Library and Information technology (IT)	7 (46%)
Research	2 (16%)
Administration and other staff	7 (38%)
Weight (kg)	86.4 (13.3)
Height (cm)	172.9 (10.4)
Body mass index, BMI (kg/m^2^)	28.1 (3.82)
Waist circumference (cm)	99 (18)
Males	99 (14.5)
Females	98.5 (NA)
Weight status	
Underweight (<18.5 kg/m^2^)	0 (0%)
Normal (18.5–24.9 kg/m^2^)	4 (29%)
Overweight (25.0–29.9 kg/m^2^)	3 (21%)
Obese (≥30.0 kg/m^2^)	7 (50%)
Blood pressure (mm Hg)	
Systolic (SBP)	129 (17)
Diastolic (DBP)	84.5 (6)
Triglycerides	2.0 (0.24)
High-density lipoprotein (HDL)	1.11 (0.29)
Low-density lipoprotein (LDL)	3.35 (0.72)
Total cholesterol	5.36 (1.02)
Framingham Risk Score (FRS)	18.3 (5.8)
Level of cardiac risk	
Moderate (10–19%)	11 (73%)
High (20% and above)	4 (27%)
Hypertensive (AOBP > 135/85 mm Hg)	
Yes	4 (29%)
No	10 (71%)

**Table 2 ijerph-20-00846-t002:** Type and frequency of initial goals set by each patient at the start of the second pharmacist-led CVD prevention program.

Category	Goal	Frequency of Reported Goal	Subgoal	Frequency of Reported Goal *
Health Behaviour				
Diet	Change Quantity	25	Reduce red meat intake	4
			Reduce processed/convenience food intake	2
			Reduce salt intake	2
			Reduce sugary food intake	3
			Reduce carbohydrate intake	4
			Increase water intake	1
			Increase vegetable intake	9
	Change Timing/Frequency	1	Eat earlier in the evening	1
	Improve Quality	9	Incorporate heart healthy foods and fats	4
			Follow diet models (ex. Mediterranean diet, Healthy plate model)	5
Physical Activity	Increase Quantity	12	Increase frequency	9
			Increase duration	3
	Change Timing (leisure-based vs. transport-based)	6	Increase intensity	4
			Increase outdoor exercise	2
Smoking	Smoking Cessation	2	Stop Cold Turkey	2
Medication use	Optimize Medication Therapy	2	Start Medication	1
			Adhere to Medication	1
Sleep	Improve Sleep Quality	1	Avoid fluids late in the evening before bed	1
			Get to bed earlier	1
Alcohol	Reduce Alcohol Intake	3	Reduce Alcohol Intake	3
Health measure				
Weight	Reduce Weight Measures	6	Reduce Weight Measures	6
Blood pressure	Reduce Blood Pressure Readings	3	Reduce Blood Pressure Readings	3
Lipids	Improve Cholesterol Levels Reduce	5	Triglyceride and LDL Cholesterol Levels	4
			Increase HDL Cholesterol Levels	1

* The number of subgoals reported may exceed the frequency of reported goals in the sample.

**Table 3 ijerph-20-00846-t003:** List of barriers to patient behavioural goals.

Goal Barriers	Description	Example
Physical (bodily) limitation	Patients report that physical limits of their body do not permit them to move forward in their pursuit of their goal (e.g., Injury, recovery from surgery, illness)	“Recovering from hip surgery”
Physical (external) limitation	Patients report that physical factors from the outside world limit their opportunities to carry out their behavioural goal activities (e.g. COVID-19 pandemic, change in weather, health provider assessment required)	“physical activity has reduced due to weather—no gardening or yard work” “limited w/COVID as cannot go to gym”
Time limitation	Patients report that time constraints limit their ability to carry out behavioural goal activities	“Getting busy with work, no time to prepare healthy snacks”
Psychological limitation (lack of motivation, cravings, biases)	Patients report that there are psychological barriers which render them unable to make progress.	“trouble getting out of bed in the morning for exercise” “Finding it somewhat difficult to stick with this, feels he is craving salt more”

## Data Availability

The data underlying this article cannot be shared publicly due to patient privacy. Data requests should be made to the corresponding author.

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
