# Peer review of "Setting Goals to Reduce Cardiovascular Risk: A Retrospective Chart Review of a Pharmacist-Led Initiative in the Workplace"

_ijerph, 2023, doi:10.3390/ijerph20010846_

Round 1

Reviewer 1 Report

This is a very well written and well referenced manuscript.  Unfortunately the sample size is too small from which to draw scientifically sound conclusions.

Author Response

Thank you for the positive feedback on the writing of our manuscript. We appreciate that there were a limited number of patients who enrolled in the second dose of the pharmacist-led CVD prevention intervention, which occurred during the COVID-19 pandemic. However, we wish to emphasise that the sample size is actually n=5413 words, since this is a qualitative-based analysis using quantitative content analysis of longitudinal data. Moreover, it is very common for qualitative study to comprise 15-20 participants. Thus, we respectfully disagree that our sample is insufficient to ‘draw scientifically sound conclusions’. Finally, we would also highlight that this research is a descriptive study to explore an area of clinical practice that is not investigated in the literature, namely patient goal-setting in CVD risk prevention.

Reviewer 2 Report

Thank you so much for giving me an excellent opportunity to review your manuscript. The manuscript is well-written, and the topic is very interesting. Cardiovascular risk reduction is a significant topic to address in public and community health. I have several comments that will be helpful for potential revision. 

Introduction
1. Research Problem: What is the research gap in this topic?
The authors stated, “A better understanding of the goal-setting will allow providers to adapt their discussions better and maintain a patient-centered approach.”  Is there any previous literature that aimed to examine the goal-setting process? It would be great to provide a research problem statement that summarizes what is already known or unknown knowledge about this topic.

2. Research Aim: The authors stated their aim as “To ‘determine’ which CV prevention goals are prioritized by patients.” Was this study determined something? Or examining or exploring the CVD goals and goal-setting process. In the abstract, the authors stated, “To ‘describe’ what do CVD prevention goals comprise of in a workplace CVD risk reduction program and how might goals change over time.” These are also not matched. I would also recommend reexamining the research aim in the introduction and abstract. Also, providing specific research questions would be helpful in understanding your study purpose.

Methods
1. Clarifying the study design may be helpful to check your
study aim and research questions, in my opinion. This study is a descriptive qualitative study using a retrospective chart review.  

2. Data collection: The authors stated, “Clinical information was unavailable in some patients.” It would be great to add some reasons why their clinical data were unavailable. 

3. Descriptive data related to goals is this study’s main data. Then, it needs to describe how these data were collected and recorded by whom (e.g., researcher staff, pharmacists, data collectors). What kinds of questions were used to ask their goal-setting? Also, any scales or instruments were used?

4. Since the sample size is 15, I think the authors provided some non-parametric statistics (e.g., median, range) in the results section. These should be noted in the data analysis section.

Results
The authors stated, “Overall, patients expressed a high degree
of confidence in achieving their goals, with two-thirds (n=10/15) reporting their confidence as 7.5 out of 10 (10=completely confidence, 0=not at all).”  It should be clearly stated what kind of scale was used and their reliability and validity (if possible).

Discussion
Implications for future research and practice can be added to the discussion. 

Thank you so much!

Author Response

Please see attachment for a point by point response to each comment. Thank you.

Reviewer 3 Report

In this study Authors have described novel CVD risk management program, which included goal-setting and medication management. 

While Authors have described in detail various risk factors and CVD-goal setting assessments. Addition of following points scan significantly add value to the designed study:

1) Apart from environmental factors, CVD risk is closely associated with genetic makeup including ethnicity, family CVD history. If data for these have been recorded, it will be great addition to the study.

2) Authors need to describe in more detail what impact occupation/profession had in CVD- risk management plan and goal setting. 

3) Including more female subjects in the study will have definitely increased impact of the study. Authors need to describe if there was any specific reason of low female participants in the study (why was this not given importance while designing the study). 

Round 2

Reviewer 2 Report

The authors tried to reflect on my comments and suggestions. 

Reviewer 3 Report

Authors have answered all my comments and made necessary changes. I recommend this manuscript for publication.